# RIPK1-Induced A1 Reactive Astrocytes in Brain in MPTP-Treated Murine Model of Parkinson’s Disease

**DOI:** 10.3390/brainsci13050733

**Published:** 2023-04-27

**Authors:** Chenmeng Qiao, Guyu Niu, Weijiang Zhao, Wei Quan, Yu Zhou, Meixuan Zhang, Ting Li, Shengyang Zhou, Wenyan Huang, Liping Zhao, Jian Wu, Chun Cui, Yanqin Shen

**Affiliations:** Department of Neurodegeneration and Injury, Wuxi School of Medicine, Jiangnan University, No. 1800, Lihu Avenue, Binhu District, Wuxi 214122, China

**Keywords:** Parkinson’s disease (PD), astrocyte, neuroinflammation, receptor-interacting protein kinase 1 (RIPK1)

## Abstract

Neuroinflammation is one of the hallmarks of Parkinson’s disease, including the massive activation of microglia and astrocytes and the release of inflammatory factors. Receptor-interacting protein kinase 1 (RIPK1) is reported to mediate cell death and inflammatory signaling, and is markedly elevated in the brain in PD mouse models. Here, we aim to explore the role of RIPK1 in regulating the neuroinflammation of PD. C57BL/6J mice were intraperitoneally injected with 1-methyl-4-phenyl-1,2,3,6-tetrahydropyridine (MPTP; 20 mg/kg four times/day), followed by necrostatin-1 treatment (Nec-1, RIPK1 inhibitor; 1.65 mg/kg once daily for seven days. Notably, the first Nec-1 was given 12 h before MPTP modeling). Behavioral tests indicated that inhibition of RIPK1 greatly relieved motor dysfunction and anxiety-like behaviors of PD mice. It also increased striatal TH expression, rescue the loss of dopaminergic neurons, and reduce activation of astrocytes in the striatum of PD mice. Furthermore, inhibition of RIPK1 expression reduced A1 astrocytes’ relative gene expression (CFB, H2-T23) and inflammatory cytokine or chemokine production (CCL2, TNF-α, IL-1β) in the striatum of PD mice. Collectively, inhibition of RIPK1 expression can provide neuroprotection to PD mice, probably through inhibition of the astrocyte A1 phenotype, and thus RIPK1 might be an important target in PD treatment.

## 1. Introduction

Parkinson’s disease (PD), the second-most common neurodegenerative disease [1], is characterized by both motor symptoms (resting tremor, delayed movement, unsteadiness, postural reflexes, etc.) and non-motor symptoms (such as gastrointestinal dysfunction and mental problems) [2]. The primary pathological features in PD include abnormal neuroinflammation, loss of dopaminergic (DA) neurons in the substantia nigra (SN), and alpha-synuclein aggregation and misfolding [3,4]. A key aspect thought to regulate PD progression is chronic brain inflammation, which is marked by persistent activation of glial cells and the subsequent sustained release of proinflammatory mediators [5].

In the process of PD, inflammatory signals induce susceptible nerve cells to undergo programmed cell death, including apoptosis and necroptosis. Receptor-interacting protein kinase 1 (RIPK1) is a multifunctional protein that belongs to the protein kinase family. It contains an N-terminal kinase domain, an RIP-like domain, and a C-terminal death domain. RIPK1 plays an important role in regulating inflammation and cell death pathways, activating the pathogen recognition receptor and causing DNA damage [6,7]. Previous reports have shown that the levels of RIPK1 are increased in brains from AD patients and a mouse model of AD (APP/PS1) [8,9]. The levels of inflammatory cytokines and memory deficits in a mouse model of AD can be reduced by using RIPK1 inhibitor Nec-1 (a highly specific inhibitor of RIPK1 activity). In addition to regulating caspase-independent necroptotic death and RIPK1-dependent apoptosis, RIPK1 can also regulate the expression of proinflammatory cytokines [10]. For example, RIPK1 mediates inflammatory signaling of microglia and astrocytes in multiple sclerosis [11] and mediates neuroinflammation caused by microglial activation in cerebral ischemic stroke [12]. However, whether RIPK1 works on neuroinflammation in PD models has not been clearly demonstrated.

In the present study, we demonstrated that RIPK1 expression was increased in the striatum of PD mice and Nec-1 can significantly protect mice against MPTP-induced neurodegeneration through inhibiting astrocyte conversation into the neurotoxic A1 type. The findings here suggest that RIPK1 inhibition could attenuate disease in MPTP-induced PD murine model by inflammation inhibition. Accordingly, these imply that therapeutic administration of a RIPK1 inhibitor may reduce the incidence of PD.

## 2. Materials and Methods

### 2.1. Animals and Treatments

Male C57BL/6J mice (seven weeks old, 18 ± 2 g) were purchased from Gem Pharmatech (Nanjing, China) and were acclimated to their environment for one week before the experiments. All mice were kept in pathogen-free environments (12 h light/12 h dark cycle) at 24 ± 2.0 °C and 55 ± 10% humidity, with free access to water and normal rodent chow meal (five mice/cage). All animal experiments were approved by the Jiangnan University Animal Ethics Committee (approval number (IACUC issue number): JN. No20211130c0800228502.

Mice were randomly divided into four groups (*n* = 10 mice/group). The control group received normal saline (the solvent of MPTP) every two hours for a total of four injections a day, and then intraperitoneally injected with normal saline containing 2% DMSO (solvent of Nec-1) for 7 days. The MPTP group received 20 mg/kg (intraperitoneal injection, i.p.) MPTP (M0896, Sigma-Aldrich, St. Louis, MO, USA) every two hours for a total of four injections a day [13], and then intraperitoneally injected with normal saline containing 2% DMSO (solvent of Nec-1) for 7 days. Notably, the first normal saline containing 2% DMSO was given 12 h before the first normal saline (for control group) or MPTP (for MPTP group) injection as control for first Nec-1 injection. The Nec-1 group received normal saline every two hours for a total of four injections a day, and then were intraperitoneally injected with 1.65 mg/kg Nec-1 (N9073, Sigma-Aldrich, St. Louis, MO, USA) solution for 7 days. The Nec-1 + MPTP group received 20 mg/kg (i.p.) MPTP every two hours for a total of four injections a day, and then were intraperitoneally injected with 1.65 mg/kg Nec-1 solution for 7 days. Notably, the first Nec-1 was given 12 h before the first MPTP injection, as reported previously [14]. The detailed experimental procedure is shown in Figure 1.

### 2.2. Behavioral Tests

Mice received behavioral pretraining once daily for three days from day 4 to day 6, as indicated in Figure 1 (green arrow). The behavioral tests were carried out on day 7 and all tests were double-blind.

**Figure 1 brainsci-13-00733-f001:**
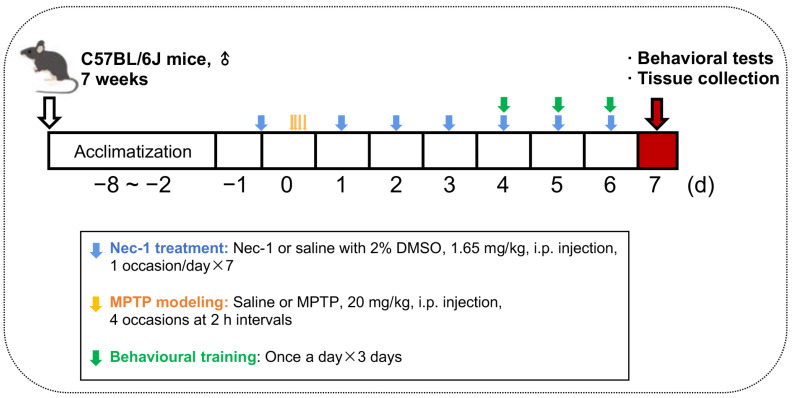
Schematic representation of the experimental procedure.

Pole descent test: The degree of bradykinesia was assessed by descent time. In detail, the home cage contained a pole that was 50 cm long, 1 cm in diameter and wrapped with non-adhesive gauze. The descent time of the mice back into their home cage was recorded once they were placed head-down on the top of the pole. Each mouse underwent the test three times (15 min intervals), with the average time being analyzed.

Open field test: An opaque plastic box (50 cm × 50 cm × 50 cm) was used as the open field here. There were nine grids (16.7 cm × 16.7 cm) on the bottom of the box. Two minutes’ video was captured while the mouse was positioned in the middle of the open field. The test sessions were captured on camera and EthoVision software was used to evaluate grid number, average speed and total distance of each mouse (Noldus, Wageningen, The Netherlands).

### 2.3. Tissue Sample Collection and Preparation

Mice were euthanized under isoflurane, deeply sedated as reported before [15]. For Western blot analysis, striatum tissue was swiftly removed from the brain. Total protein was extracted from striatum tissue by ultrasonically homogenizing 20 mg of tissue in 200 μL of RIPA buffer (P0013B, Beyotime, Shanghai, China) with 2 μL of phenylmethanesulfonyl fluoride (ST506, Beyotime, China) and 4 μL of phosphatase inhibitor (P1081, Beyotime, China). The homogenate was then centrifuged at 13,000 rpm for 5 min at 4 °C, and the supernatant was collected. The protein content was measured with a BCA protein assay kit (BL521A, Biosharp, Guangzhou, China). The obtained proteins were stored at −80 °C for subsequent detections.

For quantitative real-time PCR (qRT-PCR) analysis, total RNA was extracted from striatal tissue using TRIzol^TM^ reagent (15596018, Invitrogen, Carlsbad, CA, USA) and transcribed into cDNA using PrimeScript^TM^ RT reagent kit (RR036A, Takara Bio, Kyoto, Japan).

For immunofluorescence (IF) labeling, brains were fixed in 4% paraformaldehyde (PFA) in 0.01 M phosphate buffer (pH 7.4) overnight at 4 °C after transcardial perfusion with cold PBS. Then, they were dehydrated in 20% and 30% sucrose, each for 24 h at 4 °C. Tissue samples were subsequently embedded in O.C.T. and subjected to cryosection at 10 μm (CM1950, Leica, Wetzlar, Germany).

### 2.4. Quantitative Real-Time PCR Analysis

Quantitative real-time PCR (qPCR) was conducted on a Light-Cycler 480 II (Light-Cycler 480 II, Roche, Basel, Switzerland) using validated primer sets obtained from Primer Bank and SYBR^®^ Premix Ex TaqTM II (RR820A, Takara Bio, Kyoto, Japan). The 2^−ΔΔCt^ technique was used to compute the relative quantity of mRNAs after normalizing gene expression to GAPDH. The primer sequences used in this study are listed in Table 1.

### 2.5. Western Blot Analysis

Western blot analysis was performed as follows. Briefly, 30 μg of total protein was used to run on 10% SDS-PAGE gel, then transferred to PVDF membranes (ISEQ00010, Millipore, Burlington, VT, USA). After blocking with 5% skim milk at room temperature for 1 h, the membranes were incubated with the appropriate primary antibodies at 4 °C overnight: rabbit anti-tyrosine hydroxylase (TH, 1:1000, AB152, Millipore, Burlington, VT, USA), rabbit anti-RIP antibody (RIPK1, 1:300, ab202985, Abcam, Cambridge, UK), mouse anti-GFAP antibody (GFAP, 1:1000, MAB360, Merck, Kenilworth, NJ, USA), goat anti-Iba-1 (Iba-1, 1:1000, ab5076, Abcam, UK) or rabbit anti-GAPDH (1:8000, 10494-1-AP, Proteintech, Rockford, IL, USA). Goat anti-rabbit IgG (1:8000, BA1054, Boster, Wuhan, China) or goat anti-mouse IgG (1:8000, BA1051, Boster, Wuhan, China) conjugated with horseradish peroxidase was used as secondary antibody. Blot bands were visualized by incubation with BeyoECL Plus (P90720, Millipore, Burlington, VT, USA) for 1 min approximately and imaged by a Gel Image System (Bio-Rad, Universal Hood III, Hercules, CA, USA). ImageJ software (NIH Image, Bethesda, MD, USA) was used to accomplish the densitometry analysis. All original Western blot images in this study are presented in Appendix A.

### 2.6. Immunofluorescence Staining

For slice collection, brain slices containing the major portion of the striatum from bregma 0.14 mm to 1.18 mm and brain slices containing the major portion of the SN from bregma −2.92 mm to −3.52 mm were collected. Each mouse brain was cut into 10 μm-thick coronal slices. For immunofluorescence staining, brain slices were then immersed in 0.01 M sodium citrate buffer (pH 6.0) for antigen retrieval at 95 °C and washed twice in PBS. Additionally, brain slices were incubated for 30 min at 37 °C in PBS containing 0.3% (*v*/*v*) Triton X-100 and 10% (*v*/*v*) goat serum. The primary antibodies rabbit anti-tyrosine hydroxylase (TH, 1:1000, AB152, Millipore, Burlington, VT, USA), mouse anti-GFAP (GFAP, 1:2000, MAB360, Merck, USA), goat anti-Iba-1 (Iba-1, 1:1000, ab5076, Abcam, UK), and rabbit anti-RIP antibody (RIPK1, 1:1000, ab202985, Abcam, UK) were incubated overnight at 4 °C. Secondary antibodies were coupled with FITC-conjugated goat anti-mouse IgG (1:1000, A0568, Beyotime, China) and Cy3-conjugated donkey anti-rabbit IgG (1:1000, A0502, Beyotime, China) or Cy3-conjugated goat anti-rabbit IgG (1:1000, A0516, Beyotime, China) for 30 min at 37 °C. Nuclei were finally stained with DAPI. Fluorescence images were acquired using an Axio Imager Z2 (Zeiss, Axio Imager Z2, Oberkochen, Germany) and quantitative analysis of fluorescence images was processed using ZEN 2.3 (blue edition). ImageJ software (NIH Image, Bethesda, MD, USA) was used to calculate the number of TH-positive cells in SN and the mean fluorescence intensity of Iba-1, GFAP or RIPK1 in striatum. Detailed information regarding the counting of TH^+^ neurons and the mean fluorescence intensity of GFAP, Iba-1 and RIPK1 is as follows: 10 μm-thick brain sections were collected from the right hemisphere of mice containing the main part of SNpc (Bregma −2.92 mm to −3.52 mm) and striatum (Bregma 1.88 mm to 0.14 mm). For each mouse, 5 brain sections of the right hemisphere were selected (with a 100 μm gap between each section) and stained with IF. The number of TH^+^ neurons in the SNpc region (the main part of SNpc is located between Bregma −2.98 mm and −3.38 mm, with a 100 μm gap between each section) and the mean fluorescence intensity of GFAP, Iba-1 and RIPK1 in the striatum region (the main part of striatum is located between Bregma 0.76 mm and 1.16 mm, with a 100 μm gap between each section) of the 5 brain sections of the right hemisphere were then calculated. The average of these 5 numbers was used for the final statistical analysis. For the calculation of Manders colocalization coefficients of RIPK1 and GFAP in the striatum, ImageJ software and Coloc 2 plugin were used. The Manders colocalization coefficients of RIPK1 and GFAP in the striatal area were first calculated for the 5 brain sections in each mouse, and the average of these 5 numbers was used for the final statistical analysis.

### 2.7. Statistical Analysis

All data are presented as means ± standard error of the mean (SEM). Differences among the four groups were statistically analyzed using one-way ANOVA and LDS post hoc analysis. Values of *p* < 0.05 were considered to indicate significance. All statistical analysis was performed with GraphPad Prism 8.0.

## 3. Results

### 3.1. Nec-1 Inhibits RIPK1 Overexpression in MPTP-Treated Murine Model of PD

Nec-1 (Figure 2A) is a small-molecule inhibitor of RIPK1, and can effectively inhibit the expression of RIPK1 in the brain by crossing the blood–brain barrier [16]. Therefore, we firstly evaluated the effect of Nec-1 on the expression of RIPK1 in the striatum of MPTP-induced PD model mice using Western blot. We found that Nec-1 inhibits the expression of RIPK1 protein in the striatum of control mice (control group vs. Nec-1 group), suggesting the efficacy of Nec-1. Further, MPTP application increased the expression of RIPK1 in the striatum, and these changes were significantly reversed by Nec-1 treatment (Figure 2B,C).

### 3.2. Inhibition of RIPK1 Improves MPTP Induced Motor and Non-Motor Dysfunctions in Mice

In the pole test, the descent time was significantly increased in the MPTP group compared to the control group. In contrast, Nec-1-treated PD mice (Nec-1 + MPTP) spent significantly less time climbing the pole, suggesting that Nec-1 treatment could efficiently ameliorate the motor deficit in MPTP induced PD model (Figure 3A). Open field tests were utilized to investigate and gauge the anxiety and curiosity of mice [17]. As shown in Figure 3B–E, the grid number, average speed and total distance of PD mice were reduced greatly compared to control mice. Notably, Nec-1 treatment significantly ameliorated anxiety-like behavior of PD mice, as indicated by more grids, faster average speed and longer distance in Figure 3E. These results indicate that inhibiting RIPK1 could efficiently ameliorate motor and non-motor dysfunctions in PD mice.

### 3.3. Inhibition of RIPK1 Rescues Dopaminergic Neurons in MPTP-Treated Murine Model of PD

The progressive loss of dopaminergic neurons within SN is a definitive pathological hallmark of PD [18]. In order to verify the neuroprotection effect of Nec-1, we characterized TH^+^ neurons in SN by IF staining and TH expression by Western blot analysis. As shown in Figure 4A,B, TH^+^ neurons were markedly reduced in MPTP-induced PD mice compared to control mice, whereas this loss was dramatically inhibited by Nec-1, in contrast to PD mice. Results from densitometric analysis of Western blot indicated that MPTP considerably reduced striatal TH expression compared to the control group and Nec-1 significantly prevented this decline (Figure 4C,D). These findings implies that Nec-1 mediates the neuroprotective effect on disease progression by attenuating the loss of dopaminergic neurons in SN and the decline in striatal TH expression in PD mice.

### 3.4. Inhibition of RIPK1 Reduces Astrocyte Activation in MPTP-Treated Murine Model of PD

Studies have shown that glial cells mediate neuroinflammation contributing to the pathogenesis of PD [19]. Given the link between neuroinflammation and neuronal loss in PD [20], we next examined signs of neuroinflammation in mice among the four groups. Results of Western blot showed a significant increase in GFAP protein expression in the striatum after MPTP administration, which was attenuated after Nec-1 treatment (Nec-1 + MPTP treatment group) (Figure 5A,B). Meanwhile, IF analysis showed the same trend with a significant activation of astrocytes in the striatum of MPTP-treated group mice, in contrast to being inhibited in Nec-1 + MPTP-treated group mice. Based on the morphological characteristics of astrocytes, the astrocytes in the striata of MPTP-treated mice exhibited such features as hypertrophy, thicker and longer dendrites, an increased number of surface protrusions, and changes in nuclear size when compared to the control group. In contrast, the astrocytes in the striata of Nec-1 + MPTP-treated mice showed long and slender protrusions when compared to the MPTP group. Thus, cellular morphology implies that astrocytes in the striatum of MPTP-treated mice were reactive [20], whereas some of the astrocytes in the Nec-1 + MPTP group had not yet transformed into reactive (Figure 5D,E). However, neither the expression of striatal Iba-1 protein nor the activation of microglia in the striatum altered significantly among the four groups (Figure 5A,C,D,F). Microglia are activated on day 1 following MPTP treatment, and actively go back to a quiescent condition on day 7 after MPTP administration [21,22]. We assumed that the time of sampling here was 8 days after MPTP injection, and microglia at this time point had returned to the resting state [20]. Accordingly, we found that RIPK1 inhibition could markedly reduce the activation of astrocytes in the MPTP-treated group.

### 3.5. RIPK1 Is a Key Mediator of Disease-Associated Astrocyte Response

Based on the results above, we thereby hypothesized that Nec-1 treatment reduces neuron loss and ameliorates astrocyte activation, probably by inhibiting RIPK1 expression of astrocyte in brains of PD mice. Hence, we further assessed the expression of RIPK1 in astrocytes by calculating the coefficient of colocalization of RIPK1 and GFAP in striata of mice from all groups. In line with results from Figure 2C, RIPK1 expression in striata of PD mice was higher than that in control mice, and Nec-1 significantly inhibited this overexpression (Figure 6A,B). Notably, Nec-1 treatment significantly decreased the higher coefficient of colocalization of RIPK1 and GFAP in striata of PD mice, implying that RIPK1 is activated in the PD brain and may be involved in regulating disease-related astrocyte function.

### 3.6. Inhibition of RIPK1 Prevents the Polarization of Astrocytes into Neurotoxic A1 Type

Previous research has identified at least two possible subtypes of reactive astrocytes, designated “A1” for proinflammatory and neurotoxic astrocytes and “A2” for anti-inflammatory and neurotrophic astrocytes [23]. These activation states are differentiated by their unique transcriptional and functional patterns, which include a propensity to promote neuronal cell death (a hallmark of A1-type astrocyte activity) [24]. We next confirmed whether RIPK1 signaling was required for the polarization of astrocytes into neurotoxic A1 type in MPTP-induced PD mice. We performed a more comprehensive analysis of astrocyte activation and inflammatory factor-related gene expression in the presence of RIPK1 inhibitors. Notably, these experiments revealed that the putative A1 genes CFB (the expression of which can promote the transformation of astrocytes into A1 phenotype by activating the complement system and promoting the formation of C3b [25]) and H2-T23 (H2-T23 is a mouse MHC-encoded gene that can present specific antigenic peptides and mediate immune responses, in the CNS, H2-T23 increases the expression of IL-1β and TNF-α, mediated proinflammatory function of astrocytes [26]) remained highly expressed in MPTP-induced PD mice compared with the control group (Figure 7A,B). The expression of A1 gene SERPING1 (the expression of which can lead to overactivation of the complement system, resulting in the transformation of astrocytes into A1 type [27]) showed a similar trend to CFB and H2-T23, but the difference was not statistically significant (Figure 7C). In contrast, the expression of A2 genes S100A10, PTX3 and EMP1 remained unchanged (Figure 7D–F). Blocking RIPK1 significantly blocked the upregulation of genes involved in the polarization of astrocytes into A1 neurotoxic cells in the striata of PD mice induced by MPTP, indicating that RIPK1 plays an important role in astrocyte activation (Nec-1 + MPTP group vs. MPTP group).

We also determined the expression of cytokines (including IL-1β, TNF-α, CCL2, IL-10 and IL-22) in the striata of mice by qRT-PCR. As expected, MPTP treatment also increased the expression of inflammatory cytokines and inflammatory chemokines, including IL-1β, TNF-α and CCL2 (Figure 7G–I). When compared to PD mice, the expression of IL-1β, TNF-α and CCL2 in Nec-1 + MPTP-treated mice was dramatically reduced (Figure 7G–I). In terms of expression of anti-inflammatory factors, Nec-1 treatment increased the expression of IL-10 and IL-22 compared to the control group (Figure 7J,K). We also found that the Nec-1 + MPTP-treated group had significantly increased expression of IL-22 (Figure 7K). IL-22 has anti-inflammatory effects when acting on astrocytes: it can promote the proliferation and antioxidant capacity of astrocytes, thereby protecting the nervous system from damage caused by inflammation and oxidative stress [28]. These results collectively suggest that blocking RIPK1 can alter the phenotype of MPTP-induced astrocytes in the striatal location of PD mice, which in turn modulates neuroinflammation.

## 4. Discussion

Previous studies have shown that RIPK1 is associated with neuroinflammation in many neurodegenerative diseases. For example, Ofengeim et al. found that RIPK1 was activated in human AD pathological samples, and pharmacological and genetic inhibition of RIPK1 could reduce Aβ burden, inflammatory cytokine expression and memory deficits in an AD mouse model [9]. Nec-1 treatment or RIPK3 deficiency could efficiently inhibit the death of oligodendrocytes, microglia-mediated inflammation as well as the degeneration of axons in SOD1^G93A^ mice [29]. In another study, Re et al. found a similar protective effect of RIPK1 on human ESC-derived motor neurons by coculturing them with iPSC-derived astrocytes from patients with amyotrophic lateral sclerosis [30]. In addition, inhibition of RIPK1 can also reduce inflammatory response and improve cognitive function in the bilateral carotid artery stenosis mouse model of chronic ischemic stroke [31]. When it comes to PD, pharmacological inhibition of necroptosis with Nec-1 significantly protects dopaminergic neuronal cell death, both in vitro and in vivo [32]. Lin et al. also found that pretreatment with Nec-1 or the knockout of the RIP3/mixed lineage kinase domain-like protein (MLKL) gene to block necroptosis pathway dramatically ameliorated PD. However, they mainly focused on RIP1/RIP3/MLKL mediating dopaminergic neuron necroptosis and did not explore the role of RIPK1 in neuroinflammation in PD mice [14]. Notably, in addition to being an important regulator of necroptosis, RIPK1 is gaining increasing attention as a key driver in promoting neuroinflammation [11,12]. Understanding the role of RIPK1 in neuroinflammation in the CNS would provide important insights into the pathogenesis of human neurodegenerative diseases. However, little is known about the role of RIPK1 in neuroinflammation of PD pathogenesis. Kim et al. found that RIPK1 regulates the activation of microglia in both lipopolysaccharide-induced neuroinflammation and MPTP-induced PD mice models, but astrocytes were not studied in their work [33]. In this study, MPTP-induced PD mice showed higher expression of striatal RIPK1 in protein levels, which is consistent with a previous report [34]. We further illustrated that RIPK1 expressed in astrocytes was significantly upregulated in response to MPTP challenge. The major findings of the current research are the observation that the RIPK1 inhibitor, Nec-1 protects in vivo against dopaminergic neuronal loss and behavioral deficits in MPTP-induced PD mice. The available evidence suggests that Nec-1 may have some protective effects in this context, which could be partly attributed to its ability to inhibit astrocyte activity.

Persistent inflammation mediated by neighboring glia as well as proinflammatory molecules underlies PD’s pathologic features [35,36,37]. The two main mediators of neuroinflammation in the CNS are microglia and astrocytes [19]. In physiological conditions, glial cells play a vital role in supporting neuronal cell survival and maintaining brain homeostasis. In pathological conditions, however, they instead establish a harmful brain environment by secreting inflammatory factors. These cells are activated, which triggers a chain reaction of inflammation, releasing potentially neurotoxic cytokines and increasing neurodegeneration [23]. The activation of microglia and astrocytes are previously reported in MPTP-induced PD mice [38]. When dopaminergic neurotoxins, such as MPTP [38] and 6-hydroxydopamine (6-OHDA) [39], alter the homeostasis of neurons, astrocytes and microglial cells, neuroinflammatory activities mediated by activated glial cell accelerate neurodegeneration. Increases in glial cell size and number, as well as the induction of morphological alterations, are indicators of glial activation [40]. In the brain regions of PD animals, reactive astrogliosis, which is characterized by elevated GFAP expression levels and hypertrophy of the cell body and its extensions, has been noticed [41]. Chou et al. indicated that there are two subtypes of reactive astrocytes A1 and A2. A1 phenotype astrocytes are proinflammatory and release inflammatory mediators, whereas A2 phenotype astrocytes are anti-inflammatory and aid in the healing of damaged cells [24]. The neuroinflammatory function of astrocytes in PD is largely determined by their A1/A2 phenotype. In this paper, we show that MPTP modeling results astrocytic activation, which further assume the A1 phenotype. Our results firstly indicated RIPK1 was localized in astrocytes of PD animal model, suggesting a potential link between RIPK1 and astrocyte-mediated neuroinflammation in PD. However, it is important to note that further experiments are needed to fully understand the complexities of neuroinflammation in PD and the role of RIPK1 in this process.

## 5. Conclusions

Collectively, our results provide evidence of overexpression of RIPK1 in MPTP-induced PD mice. Under neuroinflammatory circumstances, the activation of astrocytes may be associated with the abnormal expression of RIPK1. Nec-1 blocks the activation of A1 reactive astrocytes, slows down the degeneration of dopaminergic neurons and plays a neuroprotective effect in PD mice, for example, motor and behavior improvement. Therefore, the strategy of RIPK1 inhibition in astrocytes may be more widely applied to the treatment of a variety of neurodegenerative diseases characterized and involving A1 reactive astrocyte activation. In this regard, further work is needed to understand the precise molecular mechanisms by which RIPK1 influences astrocyte immune responses.

## Figures and Tables

**Figure 2 brainsci-13-00733-f002:**
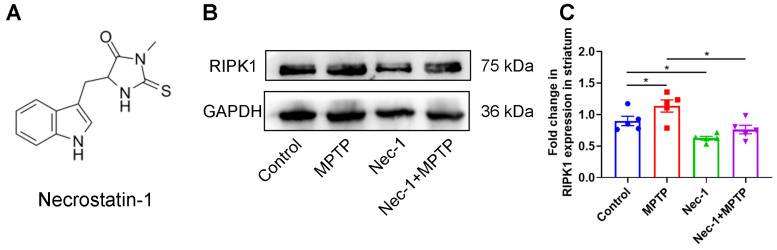
Nec-1 inhibits RIPK1 overexpression in MPTP-treated murine model of PD. (**A**) Chemical structure of Nec-1. (**B**) Representative Western blot images of striatal RIPK1. (**C**) Statistical analysis of the ratio of RIPK1/GAPDH in the striatum (*n* = 5). Statistical comparison by one-way ANOVA with post hoc comparisons of LSD. Data are expressed as means ± SEM. * *p* < 0.05.

**Figure 3 brainsci-13-00733-f003:**
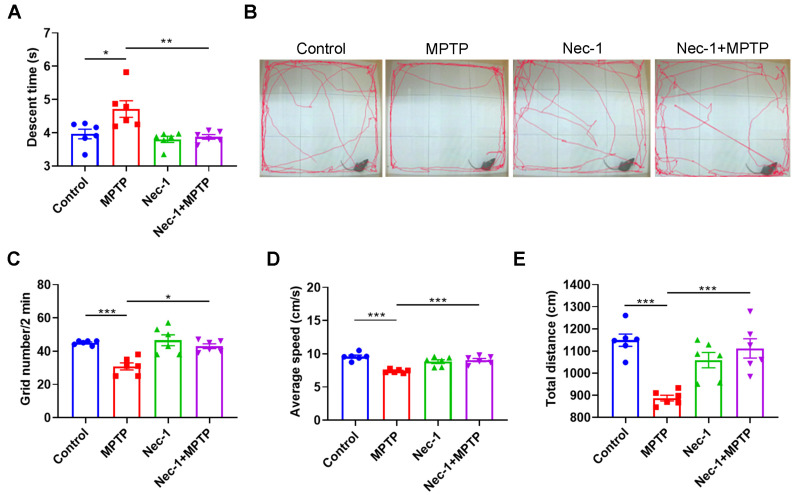
Inhibition of RIPK1 improves MPTP-treated motor and non-motor dysfunctions in mice. (**A**) Descent time in the pole test (*n* = 6). (**B**) Representative images of the open field test. (**C**) Grid number in the open field test (*n* = 6). (**D**) Average speed in the open field test. (**E**) Total distance in the open field test (*n* = 6). Statistical comparison by one-way ANOVA with post hoc comparisons of LSD. Data are expressed as means ± SEM. * *p* < 0.05, ** *p* < 0.01, *** *p* < 0.001.

**Figure 4 brainsci-13-00733-f004:**
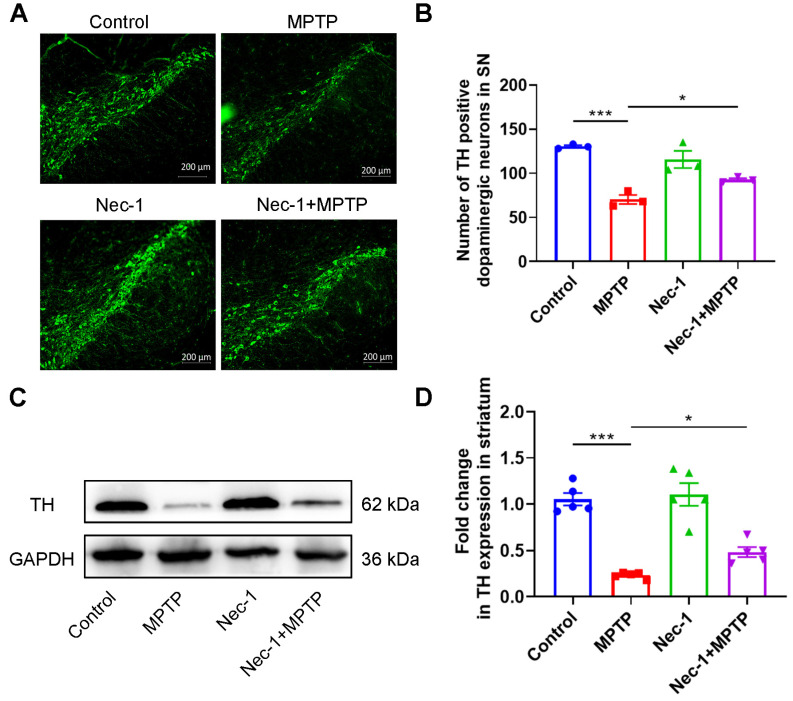
Inhibition of RIPK1 rescues dopaminergic neurons in MPTP-treated murine model of PD. (**A**) Representative IF staining for TH (dopaminergic neuron marker) in the SN. Scale bar is 200 μm. (**B**) Quantitative analysis of the number of TH-positive cells in the SN (*n* = 3). (**C**) Representative Western blot images of striatal TH. (**D**) Statistical analysis of the ratio of TH/GAPDH in the striatum (*n* = 5). Statistical comparison by one-way ANOVA with post hoc comparisons of LSD. Data are expressed as means ± SEM. * *p* < 0.05, *** *p* < 0.001.

**Figure 5 brainsci-13-00733-f005:**
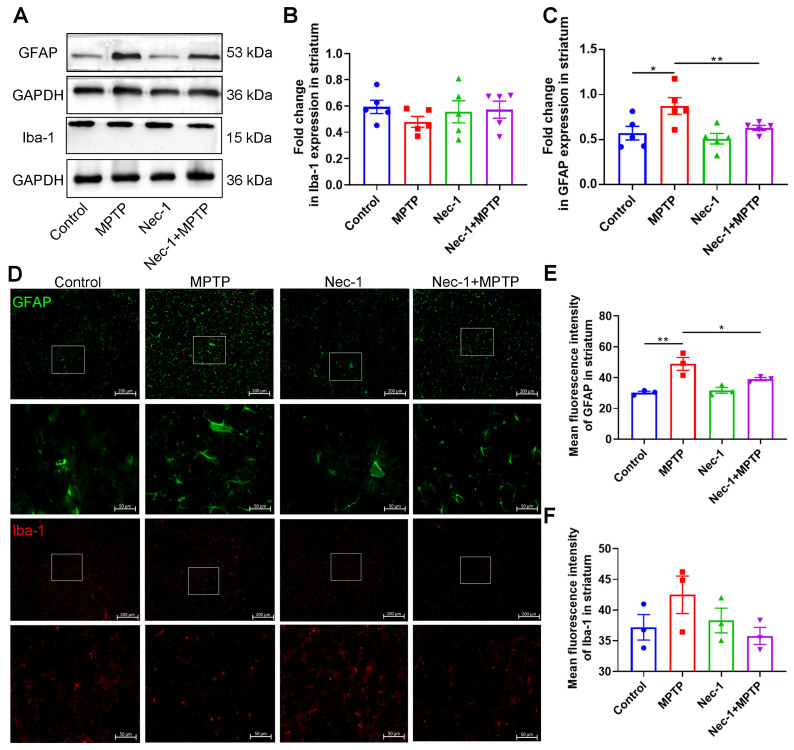
Inhibition of RIPK1 reduces astrocyte activation in MPTP-treated murine model of PD. (**A**) Representative Western blot images of striatal GFAP and Iba-1 protein. (**B**) Statistical analysis of the ratio of GFAP/GAPDH in the striatum (*n* = 5). (**C**) Statistical analysis of the ratio of Iba-1/GAPDH in the striatum (*n* = 5). (**D**) Representative IF staining for GFAP (astrocyte marker; green) and Iba-1 (microglia marker; red) in the striatum. Scale bar is 200 μm. Higher magnification views beyond the white box area are below the corresponding image. Scale bars 50 μm. Quantitative analysis of the mean fluorescence intensity of (**E**) GFAP and (**F**) Iba-1 in each group (*n* = 3). Statistical comparison by one-way ANOVA with post hoc comparisons of LSD. Data are expressed as means ± SEM. * *p* < 0.05, ** *p* < 0.01.

**Figure 6 brainsci-13-00733-f006:**
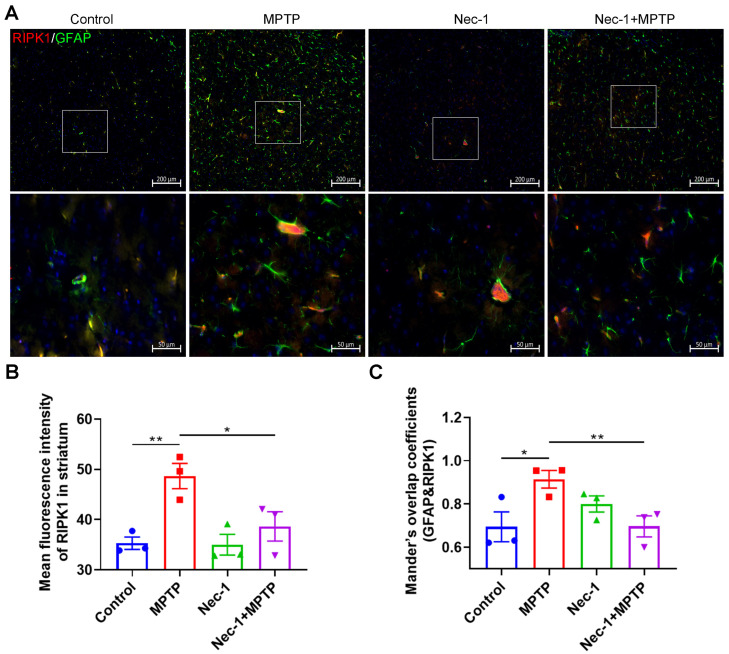
RIPK1 is a key mediator of disease-associated astrocyte response. (**A**) Representative double IF staining images for GFAP (green) and RIPK1 (red) in the striatum. Scale bar is 200 μm. Higher-magnification views in the white box area are below the corresponding image. Scale bars 50 μm. (**B**) Quantitative analysis of the mean fluorescence density of RIPK1 in each group (*n* = 3). (**C**) Quantification of Manders overlap coefficients of RIPK1 and GFAP (*n* = 3). Statistical comparison by one-way ANOVA with post hoc comparisons of LSD. Data are expressed as means ± SEM. * *p* < 0.05, ** *p* < 0.01.

**Figure 7 brainsci-13-00733-f007:**
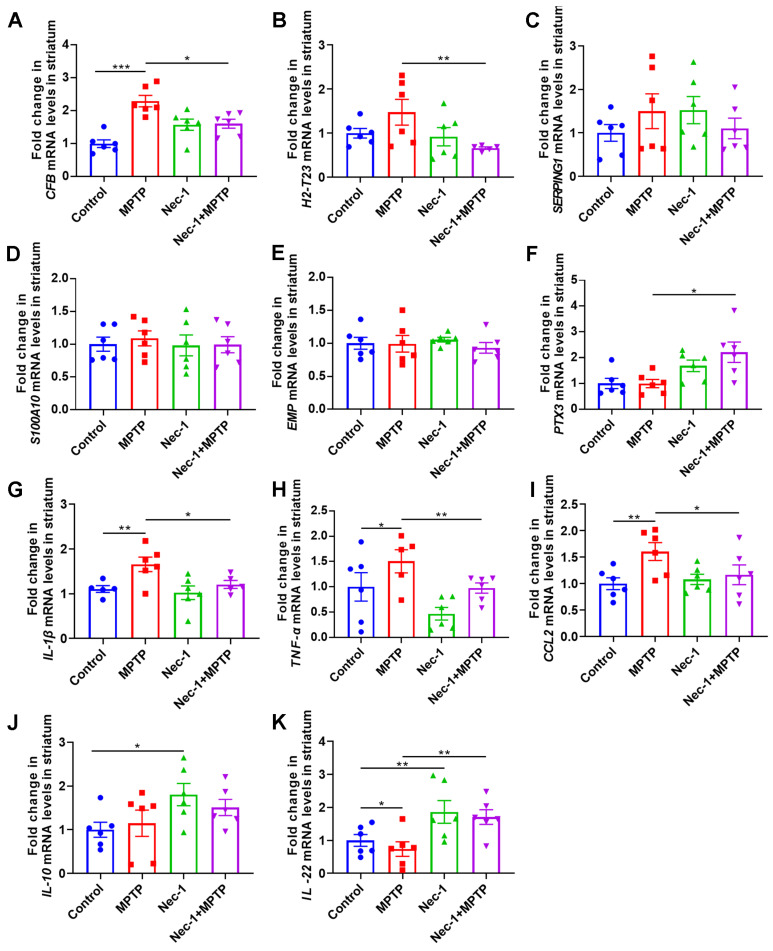
Inhibition of RIPK1 prevents the polarization of astrocytes into neurotoxic A1 type. Levels of A1-type reactive astrocyte markers (**A**) CFB, (**B**) H2-T23 and (**C**) SERPING1 in the striatum by qRT-PCR (*n* = 5–6). Levels of A2-type reactive astrocytes markers (**D**) S100A10, (**E**) EMP1 and (**F**) PTX3 in the striatum by qRT-PCR (*n* = 6). Levels of inflammatory cytokines and inflammatory chemokines (**G**) IL-1β, (**H**) TNF-α, (**I**) CCL2, (**J**) IL-10 and (**K**) IL-22 in striatum by qRT-PCR (*n* = 5–6). Statistical comparison by one-way ANOVA with post hoc comparisons of LSD. Data are expressed as means ± SEM. * *p* < 0.05, ** *p* < 0.01, *** *p* < 0.001.

**Table 1 brainsci-13-00733-t001:** Sequences of primers in qRT-PCR.

Genes	Forward and Reverse Sequences
GAPDH	Forward: 5′-AGGTCGGTGTGAACGGATTTG-3′ Reverse: 5′-TGTAGACCATGTAGTTGAGGTCA-3′
CFB	Forward: 5′-TGCTATGATGGTTACGTTCTCCG-3′ Reverse: 5′-TCCCAATAGGAATACCGGGATT-3′
H2-T23	Forward: 5′-AGAGTAACGACGAATCTCACACG-3′ Reverse: 5′-CTTGCAGGTATGCCCTCTGTT-3′
SERPING1	Forward: 5′-ATCCAAAGGTGTCACTTCTGTG-3′ Reverse: 5′-GCGGATCTTATGGTTGGTGTTC-3′
S100A10	Forward: 5′-TGGAAACCATGATGCTTACGTT-3′ Reverse: 5′-GAAGCCCACTTTGCCATCTC-3′
EMP1	Forward: 5′-TGAAGATGCTATCAAGGCAGTG-3′ Reverse: 5′-CTGGAACACGAAGACCACAAG-3′
PTX3	Forward: 5′-CGCAGGTTGTGAAACAGCAAT-3′ Reverse: 5′-GGGTTCCACTTTGTGCCATAAG-3′
IL-1β	Forward: 5′-GAAATGCCACCTTTTGACAGTG-3′ Reverse: 5′-TGGATGCTCTCATCAGGACAG-3′
TNF-α	Forward: 5′-CCTGTAGCCCACGTCGTAG-3′ Reverse: 5′-GGGAGTAGACAAGGTACAACCC-3′
CCL2	Forward: 5′-TTAAAAACCTGGATCGGAACCAA-3′ Reverse: 5′-GCATTAGCTTCAGATTTACGGGT-3′
IL-10	Forward: 5′-GCTGGACAACATACTGCTAACC-3′ Reverse: 5′-ATTTCCGATAAGGCTTGGCAA-3′
IL-22	Forward: 5′-ATGAGTTTTTCCCTTATGGGGAC-3′ Reverse: 5′-GCTGGAAGTTGGACACCTCAA-3′

## Data Availability

The data sets used and analyzed during the current study are available.

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
