# Peer review of "RIPK1-Induced A1 Reactive Astrocytes in Brain in MPTP-Treated Murine Model of Parkinson’s Disease"

_brainsci, 2023, doi:10.3390/brainsci13050733_

Round 1

Reviewer 1 Report

Summary:

Qiao et al. investigated the role of RIPK1 in regulating neuroinflammation in Parkinson's disease (PD). The researchers used a mouse model of PD induced by MPTP and treated the mice with Nec-1, an inhibitor of RIPK1. The results showed that inhibiting RIPK1 improved motor function, reduced anxiety-like behaviors, and increased the expression of genes related to dopamine in the mice. It also reduced the activation of astrocytes and inflammatory cytokine production. These findings suggest that inhibiting RIPK1 may be an effective therapeutic strategy for PD, as it can provide neuroprotection against potential astrocyte-mediated neuroinflammation.

General comments:

In general, authors provided a new angle for tackling neuroinflammation in PD, introducing the role pf RIPK1 in astrocyte activation. This could provide a new strategy for treating PD with an existing drug. The work is well supported, and the evidence laid out quite clearly. A few points that I felt would need clarification and some strong claims would need to be softened to be considered for publication in Brain Sciences.

One major point is the relation among RIPK1, Nec1 and PD has also been reported in a paper published earlier this year (Kim et al., RIPK1 Regulates Microglial Activation in Lipopolysaccharide Induced Neuroinflammation and MPTP-Induced Parkinson’s Disease Mouse Models). The researchers would need to cite this paper and also described the novelty more clearly.   

Specific comments:

In the Nec-1 rescue experiment, there is a session of Nec-1 treatment before the MPTP-induction. The authors mentioned it has been reported in a previous paper without much explanation of its rationale. Please elaborate more on why this session is necessary.

For the profiling of astrocyte as well as RIPK1 in Figure 5-7, authors should also perform similar analysis in the SN.

The authors stated that ‘inhibition of RIPK1 expression provides neuroprotection against astrocyte-mediated neuroinflammation in PD mice’, however, not enough evidence was provided to show the direct link between neuroprotection and astrocyte phenotype the researchers observed. Especially given Nec-1 has been known as an inhibitor for necroptosis, indicating it may have direct impact on the neuron rather than undirect impact via astrocyte. Authors should tone down a bit on this statement or provide extra evidence to support this claim.

A general comment on the data presentation, authors should consider overlaying individual data points on their bar graph for easier interpretation of the data from the readers.

Reviewer 2 Report

1) I would like to see original westerns in the form of supplementary

2) Table 1 The sequences of primers in qRT-PCR. Not all primers that the authors used in the work are listed here.

3) For Figure 4, it would be appropriate to provide staining for an astrocytic marker, such as GFAP. I would like to see the ratio of neurons and astrocytes. In addition, immunocytochemical staining of astrocytes with GFAP may also reflect activation of reactive astrogliosis. In general, it would be good for the authors to perform a morphological analysis of astrocytes after staining with GFAP.

4) Image J software (NIH Image, Bethesda, MD) was used to calculate the number of TH-positive cells in SN and the mean fluorescence intensity of Iba-1, GFAP or RIPK1 in striatum. With the help of which plug-in and what was the algorithm for the correlation calculation of fluorescence units. Should be described in more detail.

5) For example, Ofengeim et al. Found…. Different font.

6) Conclusions should not include references to the results of the work of other authors, but reflect the results of the study itself. The conclusion should strictly reflect the results obtained. I ask the authors to write a conclusion in accordance with my remarks

Round 2

Reviewer 2 Report

The authors took into account all my comments.